# Socio-Psychological Safety of Schools in the Context of the Settlement Type and Socio-Economic Status of the Region

**DOI:** 10.3390/bs9120139

**Published:** 2019-12-05

**Authors:** Elvira N. Gilemkhanova

**Affiliations:** Department of Educational Psychology, Institute of Psychology and Pedagogy, Kazan (Volga region) Federal University, Kazan 420008, Russia; enkazan@mail.ru

**Keywords:** socio-psychological safety, settlement type, socio-economic status

## Abstract

Under current conditions, the scientific request for the study of both internal and external factors of socio-psychological safety becomes important. In the literature there are very contradictory data on the role of contextual factors in ensuring the socio-psychological safety of schools. In our work, we consider the role of socio-economic and geographical conditions in ensuring the socio-psychological safety of the educational environment of the school from the standpoint of environmental psychology. Research questions: How does the social and economic status of the region affect the subject level of the school’s socio-psychological safety? How does the type of settlement affect the personal level of the school’s socio-psychological safety? The economic, social, and geographical status of the region affects the socio-psychological safety of the school. Therefore, contextual factors influence, but do not determine the socio-psychological safety of the educational environment of the school. Multilevel approaches, which are intended for individual, psychosocial, and contextual factors, can contribute to the promotion of the theme of the socio-psychological safety of the school.

## 1. Introduction

The school as a sociocultural phenomenon has undergone a colossal transformation associated with changes in the conditions of socialization of pupils. The growth of deviations in the child and adolescent population influenced the social status of the school and the specificity of the tasks it solved. Under current conditions, the scientific request for the study of both internal and external factors of socio-psychological security becomes important. The starting construct for this study is the concept of the school’s socio-psychological safety, which is being referred to as the backbone characteristic quality of educational environment. The socio-psychological safety of the school is regarded here as a system-forming characteristic of the effectiveness of interaction between the personality and contextual components of the educational environment. The result of this interaction is to achieve consistency, or dynamic equilibrium, of goals and values (and meanings) between personality and sociocultural environment. Therefore, the educational environment is defined as a space for the organization of such interaction. Thus, socio-psychological safety of the educational environment is described by the degree of matching of individual and contextual levels of the educational environment within the “personality—sociocultural environment” open dynamic system. It is important to note that the sociocultural environment of a school is defined as a set of psychological, pedagogical, socioeconomic, geographical, informational conditions of the education, allowing the personality to realize his goals considering the opportunities presented within the educational system.

In our work, we consider the role of socio-economic and geographical conditions in ensuring the socio-psychological safety of the educational environment of the school from the standpoint of environmental psychology. As indicators of context variables we use wages, employment, and type of the settlement (city, urban-type settlement, or village) where the studied schools are located. Our previous study [1] shows the relationship of these indicators with the proportion of children at risk for deviating behaviour at the municipal region level. The primary role of the environment is postulated in Henry’s domino theory, which is a theoretical construct for many studies of school safety [2]. According to this theory a series of five steps lead to an accident, the first step, according to Henry, is the social environment. Therefore, Kisfalusi [3] also showed that the academic performance and socio-economic status of pupils has a significant impact on the level of social and psychological vulnerability of pupils. However, the number of authors including Hall and Chapman [4] believe that there are no fundamental differences in the riskiness of the educational space, fixing the social level of this problem. Obtaining different results emphasizes the importance of conceptual empirical research in the field of the educational environment and the analysis of factors influencing what Anderson pointed out as early as the 1980s [5]. However, the methodological complexity of studying the impact is explained by several controversial issues. Argyris underlines that the object of the study is complex. Studying human behavior in schools, as in any organization, involves “ordering and conceptualizing a buzzing confusion of simultaneously existing, multilevel, mutually interacting variables” [6] (p. 501). As a result, an important methodological problem of studying the influence of various factors on the educational environment is involved, this problem is related to the choice of the level of research and the variables corresponding to this level. A serious methodological difficulty lies in distinguishing contextual and individual characteristics in the study of the educational environment. However, some scientists, in particular Brookover et al. [7], note that the ineffectiveness of previously undertaken empirical studies of the educational environment is due to unconstructive models, inadequate research tools, a small number of variables, or incorrectly chosen parameters. Coleman [8] mentions one-sidedness of the selected factors. The main difficulties, according to the author, lie in the methodologically explained choice of the causal model. According to Snow, models provide “a possible system of relationships among phenomena understood in verbal, material, graphic, or symbolic terms” [9] (p. 81). Model definition is the most crucial step for significant research. One reason for the failure in finding the impact of contextual factors on the school is focusing on finding the relationship between variables, rather than determining the mechanisms causing these relationships. This is how the model providing a comprehension of the organization of the educational environment should be designed [10]. The diagnosed variables, study design, and analysis data should flow logically from the proposed model. As pointed out by Anderson [5], the different models will require not only different variables (depending on the basis of the theory), but also different statistical technical requirements that can lead to different independent conclusions.

There are three main variants of models: Cumulative, mediated, interactive [5].

Among the most commonly used models are cumulative models. Large-scale reviews of regression analysis tend to work within the framework of this model, if each variable included in the equation is independent. Mediated models include latent variables in the analysis. Many researchers used this model in a simple way, suggesting that the environment (families, schools, and communities) affects the success of a student by first affecting relationships (parents, teacher, and peers) [5]. Relationships affect the student’s self-perception, which finally and directly affects success in interactive models. Many researchers believe that the influence on the school climate and academic success of pupils is not one-sided. They offer a model in which all variables serve as dependent and independent at the same time. Levin criticized the models, in which the “explanatory variables affect the level of student success as well, but the success of the student is supposed to have no effect on the explanatory variables” [10] (p. 275). Other researches proposed an interactive model of causation, in which abilities, motivation, and attitudes of the student, on the one hand, affect the family and peers, on the other hand, are influenced by them [5]. Although the “simultaneous operation of effects” (interactive) model is more reflective and closer to reality, it is much more difficult, because each variable must be defined in relation to any variable [10].

To select the appropriate model, it is important to determine the position regarding the degree of environmental influence on the socio-psychological safety of the educational environment and how the holistic system produces the resulting effects on pupils. In essence, the socio-psychological safety of the educational environment of the school serves as an intermediary variable between (1) the collective influence of the environment and the individual background characteristics (2) the results of the student. At the same time, in the context of studying the influence of socio-economic and geographical factors, we will consider socio-psychological safety only as a dependent variable in the framework of an aggregate model using variance analysis.

### 1.1. Research Background

With the development of science, more and more branches are emerging, the coverage of scientific problems and the proposed solutions are becoming very private and do not take into account information from other industries or fields, which reduces the possibility of applying the results for practical purposes. This leads to an increased demand for interdisciplinary integrated research. The basis for understanding the role of the educational environment in the student’s development is the idea of L. Vygotsky that mental development is mediated by the sociocultural environment [11] and should be considered in the context of “person-environment.” The need of connection of the behavior of the person and his environment in one system is also the central idea of the works of R. Barker [12], who is the founder of psychology of the environment.

This relates to K. Levin’s [13] important scientific postulates, which provided a shift in emphasis from the “nature of an object” to an analysis of a person’s relationship with his environment. In determining the need for scientific study of the mutual accommodation of a person and his environment, the developer of an interdisciplinary approach to socialization, American psychologist W. Bronfenbrenner [14] (p. 188) stressed that “this process is influenced by relationships within a given environment, as well as by the broader context in which that environment is included”.

The implementation of these ideas within the framework of the socio-psychological safety of the educational environment has been carried out in a number of studies, including the study of W. Hu, and R. Wang [15], who studied the relationship between the socio-economic diversity of schools and the academic performance of students, Olsson and Fritzell [16], who studied the relationship between the socio-economic situation and the intensity of alcohol and drug use by pupils. Increasing socio-economic stratification in schools has updated the need for an expanded study of socio-psychological phenomena (such as imitation, social comparison, comparative effect of school contexts) affecting students’ problematic behavior [17,18]. Our research is also in line with these scientific studies.

### 1.2. Research Methodology

The methodological basis of the study is the ecological approach by J. Gibson [19], which allows us to consider deviations through the prism of disturbed interactions of subsystem elements of the educational environment “personality”—“the sociocultural environment of the school”. In order to analyze the socio-psychological safety, it is necessary to study how the features of the personality are interconnected with the features of the socio-cultural environment of the school. At the same time, the study of the psychosocial characteristics of the pupils can be characterized as internal factors of the socio-psychological safety of the school and the characteristics of the socio-cultural environment of the school as the external or contextual level of the system. Let us turn to the analysis of the socio-economic and geographical status of the region as the most important external factors of educational process that determines the range of possibilities for implementation of educational purposes by pupils.

In order to achieve reliable results of the study, we take advantage of the recommendations of Anderson and used the emission method and stratification of the research sample and the emission method (exemplary cases). The use of outliers or model schools is a version of stratification, which was often recommended [10,20]. The emission method allows to maximize the difference in contextual factors, of all the 36 schools studied, the study specifically includes schools presenting alternatives, that is, those with the highest or lowest economic or social parameters of region, where schools were situated. The study specifically includes schools which were presenting different types of settlement (city, urban-type settlement, or village). Stratified samples allow to study the relationship of school variables to the result for pupils in the conditions of differences in the given parameters excluding from consideration variables that usually correlate with the school context. Brookover et al. [7] conducted research only about “alternative” schools selected samples among high and low reaching schools. The data of such studies indicate the influence of contextual factors, in particular the socio-economic level of pupils. McDill et al. [21] proposed, therefore, that if stratification is done, it should include the average groups on that dimension, as well as the extreme groups. Moreover, a special attention was paid to the characteristics of the source data, making it possible to carry out a regression analysis (normality, homogeneity of differences, and independence of variables). Therefore, according to the research model, the study took place in schools that were specifically selected for particular requirements.

### 1.3. Research Questions

How does the social and economic status of the region affect the subject level of the school’s socio-psychological safety? How does the type of settlement affect the subject level of the school’s socio-psychological safety?

## 2. Materials and Methods

Thirty-six schools were included in the study, which were localized in nine different territorial-administrative units of Republic Tatarstan (with the highest or lowest economic or social parameters of region, where schools were situated and different types of settlement) with a total enrollment of three, 232 pupils aged from 10 to 17 years.

Statistical data processing was carried out using the SPSS Statistics 21 software. Statistical analysis of the data was carried out using the main effects ANOVA analyses.

### 2.1. Participants

The distribution of data reported by participants is presented in Table 1.

### 2.2. Data Source

The study applies the following factors: Social, economic, geographical. The variables included average salary in the rubles survey item, the percentage of the employed people in the region survey item, type of the settlement (Table 2). The SES variable was average salary in school region (from 21,790 rub to 42,852 rub) and unemployment rate in the region (from 0.38 to 0.86). Information about contextual factors were obtained on the republican portal of the Ministry of Education and Science and on the website of the Federal State Statistics Service (https://gks.ru/dbscripts/munst/munst92/DBInet.cgi#1).

Whether the student was growing up in a city was coded as three; the student was growing up in a village urban type was coded as two; the student was growing up in a village was coded as one.

The outcome variable included the pupil’s sociocultural safety index. It was investigated by the author’s technique “Adolescence socio-cultural safety index” [22] (Table 2). The feelings of safety were measured with respect to different specific situations in school. The resultant questionnaire comprised 35 multiple response items rated on a 4-point Likertetype scale ranging from 1 (never) to 4 (always). The questionnaire continues with questions related to the topic of the self-relation and self-esteem in interaction with peers, self-accusation. The total result for our study were assessed in points. Statistical limits of norm from six up to 14 points.

### 2.3. Procedure

The study was conducted in 2017 personally by the author using printed questionnaires. Participants anonymously and voluntarily completed the questionnaires in the classroom during a regular class period (55 min). The class tutor was on hand at all times to explain the study objectives and planned outcomes and gave instructions on how to correctly complete the questionnaire. Participants were required to use approximately 25 min to complete the questionnaire. Data from the questionnaires were then transferred to the exсel database where they were first processed. Statistical processing of the study results was completed in 2018.

## 3. Results

Analysis of the level of wages in relation to the severity of the socio-psychological safety risk index shows a significant influence of the economic factor (Table 3). In schools located in areas with high and average earnings, the riskiness of the educational environment is significantly higher. This is amply demonstrated in Figure 1.

Studying the role of the social factor, which was assessed by the share of the employed population in the region, showed that schools located in areas with both high and low employment have an increased risk of educational environment compared to areas with average indicators of the level of employment (Table 4). The discrepancy in the groups in terms of socio-psychological safety risk index is shown in Figure 2. Obviously, the risk-taking mechanisms of the educational environment are different in areas with high and low employment, and their differentiation may be the subject of further scientific research.

Estimation of the cumulative impact of wage levels and employment levels shows that in the model, only wage rates have a significant effect, while the level of employment has a very low weight. It is important to note that economic and social indicators are weakly interrelated (Table 5.). Between the average wage and the share of the employed population in the area where the studied schools are located, the correlation was *r* = 0.26; *p* < 0.05.

Analysis of the role of the type of settlement on the socio-psychological riskiness of the educational environment shows that the level of urbanization increases the risk of the socio-psychological safety of the educational environment of the school. In cities and towns of urban type, the severity of the risk of the socio-psychological safety of the school’s educational environment is significantly higher than in villages (Table 6). At the same time, as shown in Figure 3, in the villages, the average risk-taking value is somewhat higher than this indicator in the city.

ANOVA analyses is shown in Table 7.

## 4. Discussion

According to our study, it was obtained that the economic indicator for the region increases the risk of a violation of the socio-psychological safety of the educational environment in the surrounding schools. However, as it was shown in the figure, this pattern is not linear. The same can be said more clearly about the role of such an indicator as employment. Schools located in areas with both high and low employment have the same risk. This is consistent with the position of a number of scientists [23,24], in particular Aslund et al., found that the impact of socio-economic indicator is non-linear. Votruba—Drzal [25] believes that the impact of the socio-economic indicator is mediated by parental emotional problems, lack of warmth, harsh discipline, and quality of home environment, which is further reflected in the problems of student behavior. In some studies, School Region SES also had no major effects with any of our results [26]. In others an interconnection between SES and behavioral problems was shown [27].

School size study is indirectly related to the type of settlement. As a rule, schools in cities are large-scale, in villages they are small. Studying the impact on pshychological climate in the classroom, depending on the size of the school McDill and Rigsby [21] found that class size had no effect on any school result. However, Duke and Perry [28] in the descriptive study of the 18 alternative high schools (based on interviews), came to a conclusion that in small schools the student behavior was better. In a study by the Department of Education in New York State, which also included observations, school size was negatively associated with academic results [5]. Morocco, using the ESES tool [29] revealed that small schools are perceived by pupils as more friendly. Our research confirms this position. According to the results we obtained in village schools, the risk of a violation of socio-psychological safety is fairly low than in urban and township ones.

## 5. Conclusions

The economic, social, and geographical status of a region affects the social and psychological safety of a school, but this influence is non-linear and not strong.In schools located in areas with high and average earnings, the risk of violation of the socio-psychological safety of the educational environment is significantly higher than in schools where the level of wages is lower than the average with other conditions being equal.Schools located in areas with both high and low employment have an increased risk profile of the educational environment compared to areas with average employment rates.In cities and towns of urban type, the severity of the risk of socio-psychological safety of the educational environment of the school is significantly higher than in the village than the average with other conditions being equal.

However, it should be noted that these contextual factors have weaker links with the socio-psychological safety index, as compared with other variables, in particular, which can be categorized as personal. The practical value of this study is that with this information it is possible to assess objectively the risks of social and psychological safety in a particular school and to implement preventive measures in time in the most tense schools in terms of psychological safety. Strengthening psychological prevention work with pupils in schools with multiple risk indicators is more appropriate.

## 6. Strengths and Limitations

The key strength of this study is that the large sample size allowed us to conduct stratified studies at the school level. We investigated the relationship at the level of the major areas that provided a big diversity in the social, economic, and geographical features.

The main limitation of our study is cross-sectional design. Having longitudinal results at the student level would increase opportunities for causal findings, but since such data is not available, we need to use cross-section information. Therefore, quantitative results specify the characteristics of socio-psychological safety at the level of contextual and individual indicators, but long-term research is required for causal interpretation of conclusions. It is also important to note that the socio-economic status of the region and the type of settlement are not the only factors affecting the safety of the educational environment. Additional variables outside the school context may have a greater impact on socio-psychological safety. It is advisable not to absolutize the results, but to take it into account, in order to strengthen measures of psychological and pedagogical support for pupils whose schools have multiple risks of socio-psychological safety.

## Figures and Tables

**Figure 1 behavsci-09-00139-f001:**
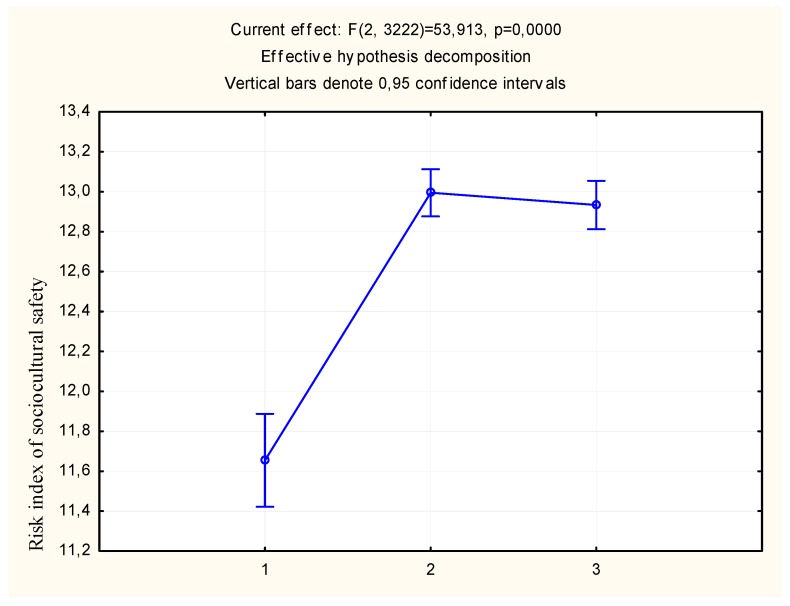
The severity of the socio-psychological safety risk index depends of the level of wages. Note: Salary (3-high level; 2-average level; 1-low level).

**Figure 2 behavsci-09-00139-f002:**
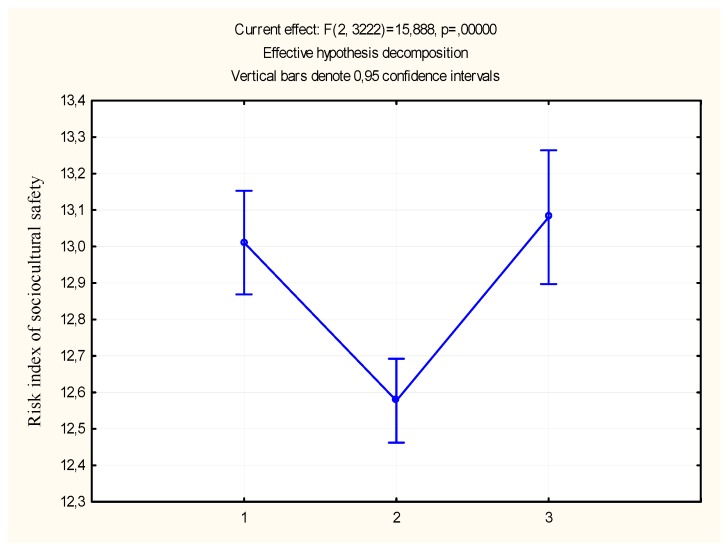
The severity of the socio-psychological safety risk index depends of the share of the employed population in the region. Note: Salary (3-high level; 2-average level; 1-low level).

**Figure 3 behavsci-09-00139-f003:**
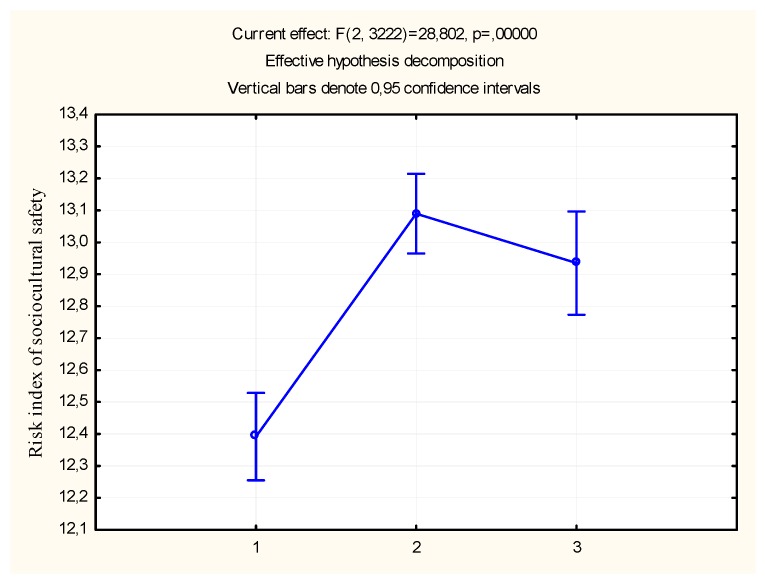
The severity of the socio-psychological safety risk index depends of the type of the settlement. Note: 3-City; 2-urban-type settlement; 1-village.

**Table 1 behavsci-09-00139-t001:** The demographic variables according to sex.

Gender		Frequency	Percentage	Valid Percentage
**Valid**	Male	1456	45	45
	Female	1618	50	50
	Not specify	158	5	5
	Total	3232	100	100

The sample is formed of 50% females and 45% males.

**Table 2 behavsci-09-00139-t002:** Descriptions of measures included in analyses.

Outcome variable Index of safety	Technique “Adolescence socio-cultural safety index”. 3375 pupils from 13 to 16 years (53% of girls, 47% of boys) participated in a research. Psychometric characteristics of a technique are provided: Reliability (α _Cr_ = 0.76), validity (r = 0.71), discriminatory power (δ = 0.9). Technique “Adolescence socio-cultural safety index” has general scale “Index of sociocultural safety” and three additional scales: Socio-psychological vulnerability; socio-psychological disintegration; virtual authorization. Socio-psychological vulnerability—characteristic of the subject of school interactions, causing high susceptibility to risks of the educational environment. It is connected with passivity, pliability to external influence, hypersensitivity to nuances of social interaction. Problems of social and psychological interaction of the pupil and other subjects of an educational system are expressed in the self-destroying trends, the direction of destructive impulses on themselves, and is followed by sense of guilt, the sensitivity increased by uneasiness. Examples of questions: For some reason as a rule all cones pour on me. If other people cause in me delight and a charm, I am disappointed in myself more and more. Socio-psychological disintegration—characteristic of the subject of school interactions, who’s problems connected with a mismatch of individual mental introject and dispositions with background characteristics of the educational environment and subjects of educational process. Key risk—isolation, the estrangement defining also inertness and not inclusiveness in an educational system. Examples of questions: I am not satisfied with those relations which at me developed with schoolmates. I would like to pass into other class. Virtual authorization characteristic of the subject of school interactions, causing social and psychological disadaptation as a result of the broken social interaction which is expressed in preference of the depersonalized communication, problems of self-identification, marginal identity and an unproductive reflection. Examples of questions: Communication is more free and entertaining on social networks. The avatar or the status incognito on social networks allows me more stoutly to prove myself, without being distracted by insignificant details.
Contextual variables	
Wages	Average amount of salary in rubles High level—16 schools Average level—16 schools Low level—4 schools
Employment	Percentage of employed people in the region High level—8 schools Average level—12 schools Low level—12 schools
Type of the settlement	Three types were identified: City, urban-type settlements, and village. In the study, the status of the city had eight settlements, 16 urban-type settlements, 12 villages.

**Table 3 behavsci-09-00139-t003:** The influence of wages to the severity of the socio-psychological safety risk index.

Effect	Sigma-Restricted Parameterization Effective Hypothesis Decomposition				
	SS	Degree of Freedom	MS	F	p
Intercept	**346,397.4**	**1**	**346,397.4**	**65,504.63**	**0.000**
Wages (3-high level; 2-average level; 1-low level)	**570.2**	**2**	**285.1**	**53.91**	**0.000**
Error	17,038.4	3222	5.3		

**Table 4 behavsci-09-00139-t004:** The influence of the share of the employed population in the region to the severity of the socio-psychological safety risk index.

Effect	Sigma-Restricted Parameterization Effective Hypothesis Decomposition				
	SS	Degree of Freedom	MS	F	p
Intercept	**464,469.6**	**1**	**464,469.6**	**85,826.37**	**0.000**
The share of the employed population in the region (3-high level; 2-average level; 1-low level)	**172.0**	**2**	**86.0**	**15.89**	**0.000**
Error	17,436.6	3222	5.4		

**Table 5 behavsci-09-00139-t005:** The influence of wages and the share of the employed population in the region to the severity of the socio-psychological safety risk index.

Effect	Sigma-Restricted Parameterization Effective Hypothesis Decomposition				
	SS	Degree of Freedom	MS	F	P
Intercept	**265,816.5**	**1**	**265,816.5**	**50,293.87**	**0.000**
Wages (3-high level; 2-average level; 1-low level)	**418.1**	**2**	**209.0**	**39.55**	**0.000**
The share of the employed population in the region (3-high level; 2-average level; 1-low level)	19.8	2	9.9	1.87	0.153541
Error	17,018.6	3220	5.3		

**Table 6 behavsci-09-00139-t006:** The influence of type of the settlement to the severity of the socio-psychological safety risk index.

Effect	Sigma-Restricted Parameterization Effective Hypothesis Decomposition				
	SS	Degree of Freedom	MS	F	p
Intercept	**504,903.4**	**1**	**504,903.4**	**94,038.47**	**0.000**
City-3; urban-type settlement-2; village-1	**309.3**	**2**	**154.6**	**28.80**	**0.000**
Error	17,299.3	3222	5.4		

**Table 7 behavsci-09-00139-t007:** ANOVA analyses.

	Standardized Coefficients		F	p
	Beta	Standard Error		
The share of the employed population in the region	−0.024	0.049	0.230	0.631
type of the settlement	0.055	0.031	3.143	0.076
Wages	0.146	0.026	32.169	0.000

The adjusted R-square is 0.04, which only explains 4% of the variability of the dependent variable.

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
