# Peer review of "Socio-Psychological Safety of Schools in the Context of the Settlement Type and Socio-Economic Status of the Region"

_behavsci, 2019, doi:10.3390/bs9120139_

Round 1

Reviewer 1 Report

The authors presented a paper reporting data partially already published years before where they consider the role of socio-economic and geographical conditions in ensuring the socio-psychological security of the educational environment of the school. They conclude that economic, social and geographical status of the region affects the social and psychological security of the school.

The methods followed by authors is fine, as well as materials including 36 schools and 3232 pupils. I wonder, in any case, what is the real contribution of the authors compared to the data already published.

In conclusion, what do they suggest with these data in their hands? What really could they suggest to improve or modify the situation based on their data. The same from a psychological point of view.

Author Response

I am very grateful to the reviewer for carefully reading my work and the important comments through which I hope that my research will become better described. Please see my revisions and explanations in the attachment. Changes (mainly additions) made in the article and response are shown in red.

Reviewer 2 Report

The topic of security in schools is one of great importance for the socio-pedagogical approach in sustainable education. However, there are aspects in the paper that need to be addressed and revised:

References for this paper need updating: the author references 80’s papers underlying the lack of research concerning the factors that influence educational environment. There is no longer the case. The introduction fails to clarify the concepts used as variables in the research. It lacks a clear research model on which to build the research design. The paper has a vague methodological approach and does not provide clear information regarding the procedure for data collection. The Instrument section is not sufficiently unambiguous when trying to present the instruments used in collecting data regarding the study’s variables. The paper tends to oversimplify the relations between demographic and environmental factors and socio-psychological security of schools. Mediation effects are not considered. Some conclusions of this paper are not fully backed by research findings, trying to overreach their interpretation. Statements such as “In schools located in areas with high and average earnings, the risk of violation of the socio-psychological security of the educational environment is significantly higher than in schools where he level of wages is lower than the average” or „In cities and towns of urban type, the severity of the risk of socio-psychological security of the educational environment of the school is significantly higher than in the village” are fairly questionable and hazardous in terms of their generalizing ability. Limitations for the study are not properly/sufficiently discussed. Citation is inconsistent and lacks standardization.

Therefore, it is recommended:

a more accurate approach in clarifying research variables and research model, diligence in using a research design and research methodology, a very thorough analysis of the third variable issue, a serious analysis of the study’s limitations, documentation should be updated, using the citation standards throughout the paper, in a systematic manner.

Author Response

(The authors gave the same response as above.)

Round 2

Reviewer 1 Report

Authors mad appropriate changes especially adding useful explanation and now the manuscript could have some advantages and utility to the readers.

Author Response

Dear reviewer! Thank you so much for helping me to improve my manuscript.

Reviewer 2 Report

The manuscript has been significantly improved.

I just have a minor remark: Research Background - I would suggest to add a comprehensive literature review on this subject. It is a lack of up-to- date reference.

Author Response

Dear reviewer! Thank you so much for helping me to improve my manuscript.

I made new corrections to the manuscript in accordance with the last observation.

Response to Reviewer 2 Comments

Point 1:

Research Background - I would suggest to add a comprehensive literature review on this subject. It is a lack of up-to- date reference.

Response 1:

I added section 1.1. Research Background to the manuscript (P. 3). The added fragment is highlighted in green.

New references to section have been included:

Levin K. Field Theory in the Social Sciences. SPb.: Speech. 2000, 365 p. Bronfenbrenner U. Ecological systems theory. Annals of Child Development. 1989, â„–6, 187-249. Hu, W.; Wang, R. Segregation in urban education: Evidence from public schools in shanghai, China. 2019, 87, 106-113. doi:10.1016/j.cities.2018.12.031 Olsson, G.; Fritzell, J. A multilevel study on ethnic and socioeconomic school stratification and health-related behaviors among students in Stockholm. Journal of School Health. 2015, 85(12), 871–879. Chen, P.; Vazsonyi, A. T. Future orientation, school contexts, and problem behaviors: A multilevel study. Journal of Youth and Adolescence. 2013, 42, 67–81. Alm, S.; Låftman, S. B. Future orientation climate in the school class: Relations to adolescent delinquency, heavy alcohol use, and internalizing problems. Children and Youth Services Review. 2016, 70, 324-331. doi:10.1016/j.childyouth.2016.09.021